# StateX: Enhancing RNN Recall via Post-training State Expansion

## Abstract

While Transformer-based models have demonstrated remarkable language modeling performance, their high complexities result in high costs when processing long contexts. In contrast, recurrent neural networks (RNNs) such as linear attention and state space models have gained popularity due to their constant per-token complexities. However, these recurrent models struggle with tasks that require accurate recall of contextual information from long contexts, because all contextual information is compressed into a constant-size recurrent state. Previous works have shown that recall ability is positively correlated with the recurrent state size, yet directly training RNNs with larger recurrent states results in high training costs. In this paper, we introduce StateX, a training pipeline for efficiently expanding the states of pre-trained RNNs through post-training. For two popular classes of RNNs, linear attention and state space models, we design post-training architectural modifications to scale up the state size with no or negligible increase in model parameters. Experiments on models up to 1.3B parameters demonstrate that StateX efficiently enhances the recall and in-context learning ability of RNNs without incurring high post-training costs or compromising other capabilities.

## 1 Introduction

Recently, recurrent neural networks (RNNs) such as gated linear attention (GLA) (Yang et al., 2024) and Mamba2 (Dao & Gu, 2024) have shown promising capabilities in language modeling. These architectures have constant per-token complexity, while the more popular Transformer architecture (Vaswani et al., 2023) has per-token complexity that grows linearly with the context length. Thus, RNNs are much more efficient than Transformers in processing long contexts.

However, RNNs still underperform Transformers in certain aspects, with one of the most critical being the long-context recall capability (Jelassi et al., 2024b). Different from Transformers, which store the representations of every token in the context, RNNs compress all contextual information into a constant-size *state*[1]. As a result, the recall ability of RNNs heavily depends on the size and capacity of this state (Jelassi et al., 2024a; Arora et al., 2024a; Yang et al., 2025; Chen et al., 2025). Despite the positive gains of increasing the state size, considering the increased training costs and the limited benefits in short-context scenarios and various downstream tasks, most RNNs are still trained with a relatively small state size compared to the rest of the model. For instance, in Mamba2-2.8B and GLA-1.3B, their recurrent states are smaller than 2% of their model sizes.

In this paper, we propose StateX, which expands the state size while keeping the training costs low and introducing little to no additional parameters. Specifically, we expand the state size of pre-trained RNNs through post-training on much less data than pre-training. Moreover, since larger recurrent states are more important for long-context models, we perform state expansion prior to long-context post-training (LPT). The training pipeline is illustrated in Figure 1.

The state expansion process is an architectural change and depends on the pre-trained model architecture. Therefore, we design two state expansion methods, targeting two popular RNN classes: linear attention (Katharopoulos et al., 2020; Yang et al., 2024) and state space models (Dao & Gu, 2024). Additionally, we explore various parameter initialization techniques and select key layers

---

[1]This is also called *recurrent state* in various contexts. We use these two terms interchangeably in this paper.

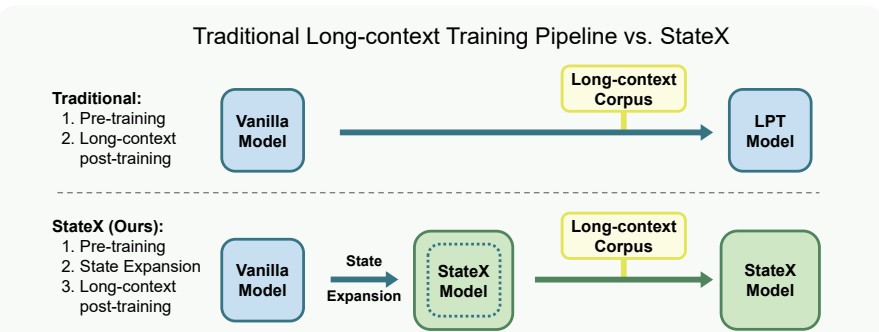

Figure 1: Difference between the traditional pipeline and StateX for training long-context models. We introduce a state expansion step (architectural modification) before the long-context post-training (LPT) stage to enhance RNN recall abilities without requiring expensive re-training.

for expansion rather than all layers, to balance model performance and adaptation efficiency. Compared to other state expansion methods that require training from scratch (e.g., MoM (Du et al., 2025), LaCT (Zhang et al., 2025)), our method is simpler and can be seamlessly applied to existing effective RNN implementations and training pipelines.

We evaluate our method on public 1.3B parameter checkpoints of GLA[2] and Mamba2[3], by conducting post-training on 10B tokens. Our empirical results demonstrate that, compared to the traditional two-stage method, StateX significantly improves performance on recall-intensive tasks, in-context learning tasks, and needle-in-a-haystack (NIAH) (Hsieh et al., 2024) tasks while maintaining performance on common-sense reasoning tasks. While using the same amount of training data as ordinary LPT, StateX yields consistently better results: the relative accuracy gain in recall-intensive tasks is 3.36% for GLA and 1.1% for Mamba2, and the relative performance gain in in-context learning is 7.2% for GLA and 1.0% for Mamba2. Also, the average NIAH accuracy up to 64K context length improves from 26.0% to 42.2% for GLA, and from 33.2% to 39.2% for Mamba2.

Overall, our contributions include:

- To the best of our knowledge, StateX represents the first work that focuses on expanding the state size of RNNs through post-training.
- For two popular RNN variants, GLA and Mamba2, we design simple and effective state expansion techniques and training recipes for efficient post-training.
- We evaluate our method on public 1.3B checkpoints. Our results show consistent improvements in recall-intensive tasks, in-context learning, and long-context retrieval, without sacrificing performance on common-sense reasoning benchmarks.

## 2 RELATED WORKS

In this section, we provide a brief description of RNNs and related work on expanding their state sizes. For more details about RNNs, please refer to the surveys (Wang et al., 2025; Lv et al., 2025).

**Modern RNNs** Recently, some RNN variants have shown promising results in sequence modeling. Some representative examples include state space models (SSMs) (Dao & Gu, 2024; Gu & Dao, 2024), the RWKV series (Peng et al., 2025; 2024; 2023), linear attention models (Katharopoulos et al., 2020; Sun et al., 2023; Yang et al., 2024), and DeltaNet (Yang et al., 2025). Some results have shown that these RNNs can outperform Transformers up to several billion parameters on certain language tasks, such as common-sense reasoning (Waleffe et al., 2024; Team, 2024), and some hybrid models have scaled up to over 100B parameters and trillions of training tokens (MiniMax et al., 2025). RNNs are attractive alternatives to Transformers because their per-token complexity is constant, while Transformers' per-token complexity scales linearly with the context length.

---

[2]https://huggingface.co/fla-hub/gla-1.3B-100B
[3]https://huggingface.co/AntonV/mamba2-1.3b-hf

| Method | Performance | Efficient Training | Easy Adoption |
|---|:---:|:---:|:---:|
| Vanilla RNNs (small states) | ✗ | ✓ | ✓ |
| Training large states from scratch | ✓ | ✗ | ✓ |
| Novel architectures with large states | ? | ? | ✗ |
| StateX (ours) | ✓ | ✓ | ✓ |

Table 1: Comparison between our work and existing approaches for increasing RNN state sizes. Vanilla RNNs underperform due to their smaller state sizes. "?" means that these works are rather new and therefore yet to be extensively tested at scale.

However, since Transformers cache all previous token representations, they outperform RNNs in recalling contextual information. This is one of the reasons why RNNs have seen limited adoption.

**Increasing RNN State Size**   Many previous works have investigated the influence of state size on the capabilities of RNNs. One important improvement of modern RNNs over previous works such as LSTM (Hochreiter & Schmidhuber, 1997) and GRU (Cho et al., 2014) is the adoption of larger matrix-valued recurrent states over smaller vector-valued states (Sun et al., 2023; Qin et al., 2024; Katharopoulos et al., 2020; Hua et al., 2022). Some later efforts focus on improving the forget mechanisms to remove unneeded information in the recurrent states, saving capacity to store more contextual information (Gu & Dao, 2024; Schlag et al., 2021). Arora et al. (2024a) provides a comprehensive comparison of the recall-throughput tradeoff of various recent RNN architectures. Although these methods show promising results, their state size is still rather small, and they lag behind Transformers in recall-intensive tasks.

**Recent State Expansion Works**   More recently, Du et al. (2025) proposes MoM, a new architecture that maintains a large state size but with lower computational overhead, by updating only parts of the recurrent state at each time step. LaCT (Zhang et al., 2025) is a concurrent work to ours that proposes a novel recurrent architecture based on the test-time training (TTT) framework (Sun et al., 2025). LaCT utilizes a much larger state than other RNNs (e.g., GLA and Mamba2) and has demonstrated strong recall and long-context capabilities. Another relevant concurrent work is by Liu et al. (2025). They utilize low-rank projections to increase the state size of RNNs with small parameter overhead, resulting in considerably better recall performance. However, these architectures have not been thoroughly evaluated across different tasks and may be hard to adopt into existing codebases.

In brief, the state size is a critical bottleneck of RNNs. Increasing the state size provides consistent performance gains for many RNN variants. However, previous works on expanding RNN states are trained from scratch, which is highly expensive and requires significant changes to the model architecture and implementation. This paper, to the best of our knowledge, is the first effort to expand states through post-training. Compared to existing architectures with larger states, our method is simpler and can be seamlessly integrated into popular RNN variants such as linear attention methods and SSMs. Table 1 shows the comparison between our work and existing works with larger states.

## 3   PRELIMINARIES

In this section, we first provide a formulation of RNNs as well as two variants—GLA and SSM (Sections 3.1, 3.2, and 3.3). Then, we discuss how the recurrent state size influences the models' recall capabilities and cost-efficiency (Section 3.4).

### 3.1   RECURRENT NEURAL NETWORKS

In RNNs, all contextual information is stored in a constant-size *recurrent state* $\mathbf{S}_t$, where $t$ denotes the time step. At each time step, new information is inserted into the previous state $\mathbf{S}_{t-1}$ with an *update rule* $f_{\text{update}}$, and then retrieves information from $\mathbf{S}_t$ with a *query rule* $f_{\text{query}}$, which is given as

$$\begin{aligned} \mathbf{S}_t &= f_{\text{update}}(\mathbf{S}_{t-1}, \mathbf{x}_t), \\ \mathbf{y}_t &= f_{\text{query}}(\mathbf{S}_t, \mathbf{x}_t), \end{aligned} \tag{1}$$

where $\mathbf{x}_t, \mathbf{y}_t \in \mathbb{R}^d$ are the input and output representations at the time step $t$. In this paper, we define *state size* as the parameter count of $\mathbf{S}_t$.

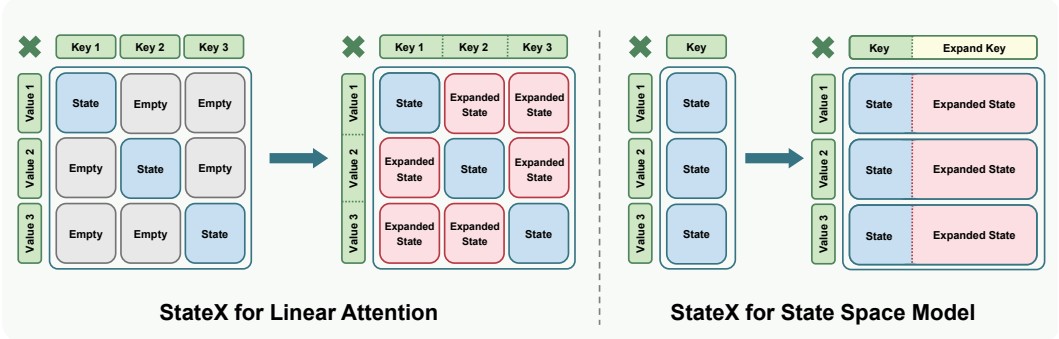

Figure 2: Illustration of StateX (our method) for expanding the state size of linear attention and state space models with little to no parameter increase. The red parts indicate the additional state parameters unlocked by StateX.

## 3.2 GATED LINEAR ATTENTION

The GLA model consists of a stack of interleaved layers of GLA blocks and feed-forward network (FFN) blocks. Since we only modify the GLA block, we omit the formulation for FFNs. Each GLA block consists of $H$ heads computed in parallel, and the layer output is the sum of the head outputs. Each GLA head can be formulated as:

$$
\begin{aligned}
\Box_{t,h} &= \mathbf{x}_t \mathbf{W}_{\Box,h}, \quad \Box \in \{\mathbf{q}, \mathbf{k}, \mathbf{v}\}, \\
\mathbf{F}_{t,h} &= \mathrm{diag}(\boldsymbol{\alpha}_{t,h}) \in \mathbb{R}^{d_k \times d_k}, \\
\mathbf{S}_{t,h} &= \mathbf{F}_{t,h} \mathbf{S}_{t-1,h} + \mathbf{k}_{t,h}^\top \mathbf{v}_{t,h} \in \mathbb{R}^{d_k \times d_v}, \\
\mathbf{y}_{t,h} &= \mathbf{q}_{t,h} \mathbf{S}_{t,h} \in \mathbb{R}^{d_v},
\end{aligned}
\tag{2}
$$

where $h \in \{1, \cdots, H\}$ is the head index, $d_k, d_v$ are the key and value dimensions. $\mathbf{x}_t, \mathbf{y}_t \in \mathbb{R}^d$ denote the input and output representations at the time step $t$, respectively, $\mathbf{q}_{t,h}, \mathbf{k}_{t,h}, \boldsymbol{\alpha}_{t,h} \in \mathbb{R}^{d_k}, \mathbf{v}_{t,h} \in \mathbb{R}^{d_v}$ are projection functions of $\mathbf{x}_t$, and LN denotes RMSNorm (Zhang & Sennrich, 2019). The state size in each GLA layer is $H d_k d_v$.

## 3.3 STATE SPACE MODELS

We focus on Mamba2, which is a state-of-the-art SSM. A Mamba2 layer can be formulated as:[4]

$$
\begin{aligned}
\mathbf{v}_{t,h} &= f_v(\mathbf{x}_t, \theta_{v,h}) \in \mathbb{R}^{d_v}, \\
\mathbf{k}_t &= f_k(\mathbf{x}_t, \theta_k) \in \mathbb{R}^{d_k}, \\
\mathbf{q}_t &= f_q(\mathbf{x}_t, \theta_q) \in \mathbb{R}^{d_k}, \\
\Delta_{t,h} &= f_\Delta(\mathbf{x}_t, \theta_{\Delta,h}) \in \mathbb{R}, \\
\alpha_{t,h} &= \exp(-\Delta_t A_h) \in \mathbb{R}, \\
\mathbf{S}_{t,h} &= \mathbf{S}_{t-1,h} \alpha_{t,h} + \Delta_{t,h} \mathbf{k}_t^\top \mathbf{v}_{t,h} \in \mathbb{R}^{d_k \times d_v}, \\
\mathbf{y}_{t,h} &= \mathbf{q}_t \mathbf{S}_{t,h} + D_h \mathbf{v}_{t,h} \in \mathbb{R}^{d_v},
\end{aligned}
\tag{3}
$$

where $f_v, f_k, f_q, f_\Delta$ are differentiable projection functions parameterized with $\theta_v, \theta_k, \theta_q, \theta_{\Delta,h}$, respectively, $A_h, D_h$ are learnable parameters. $d_k$ and $d_v$ are hyperparameters and are called the *state dimension* and *head dimension* in SSM literature. The state size of Mamba2 is also $H d_k d_v$, although these hyperparameter values may differ from GLA.

**Relationship with GLA** It has been identified that Mamba2 can be viewed as a variant of GLA (Yang et al., 2024) where heads share the same query/key (QK). In this paper, we view these two variants as different because this QK sharing mechanism influences our state expansion.

---

[4]We use attention notations ($\mathbf{q}_t, \mathbf{k}_t, \mathbf{v}_t$) instead of SSM notations ($x_t, B_t, C_t$) from the Mamba2 paper for simplicity and to highlight the analogy between the two RNN variants.

## 3.4 Influence of State Size

**Recall Ability** Since all contextual information is stored in $\mathbf{S}_t$, the ability of RNNs to recall contextual information depends on the capacity of $\mathbf{S}_t$, which in turn depends on the size of $\mathbf{S}_t$. Extensive empirical evidence indicates a strong positive correlation between the size of the recurrent states and their performance on recall-intensive tasks (Arora et al., 2024a; Hua et al., 2022; Zhang et al., 2025; Jelassi et al., 2024b). These findings highlight the critical role of state size in determining RNN recall abilities, underscoring the importance of state expansion for improving recall capabilities.

**Efficiency** The computational complexity of the token mixing component (i.e., update rule and query rule) scales linearly with the state size. Therefore, blindly increasing the state size can lead to high training and inference costs. StateX alleviates these problems during both training and inference by expanding the states via post-training (so the model is trained with smaller states most of the time) and expanding only a subset of layers.

## 4 Method

Our method, StateX, involves architectural modifications that expand the RNN state sizes prior to long-context post-training to boost their recall abilities. Meanwhile, we aim to minimize the additional parameters introduced by this modification and keep the final architecture similar to the original architecture to make it easier for the modified model to adapt. An overview of the architectural modifications is illustrated in Figure 2.

In this section, we describe the state expansion recipe for two popular classes of RNNs—GLA (Yang et al., 2024) and SSM (Dao & Gu, 2024) (Sections 4.1 and 4.2). Then, we describe parameter initialization methods after the expansion (Section 4.3) and which layers to expand (Section 4.4).

### 4.1 StateX for GLA

Since GLA employs a multi-head mechanism with different query, key, and value (QKV) vectors for each head, we can increase the state size by simply merging multiple heads into one larger head. This is because the state size of $H$ heads is $H \times d_k \times d_v$, and merging them into one head results in a state size of $1 \times Hd_k \times Hd_v$, which is $H$ times larger. Meanwhile, no additional parameters are introduced since the total number of channels in the QKV vectors remains the same. The effect of this change is illustrated in the left side of Figure 2. Merging GLA heads activates non-diagonal regions of the state matrix, thereby achieving larger states than the multi-head counterparts.

In implementation, the only difference between GLA with expanded states and the vanilla formulation (described in Section 3.2) is the number of heads and head dimension. Thus, this modification can be seamlessly applied to existing GLA implementations. We always merge all heads into one large head. This is motivated by the finding that single-head GLA generally outperforms multi-head GLA (reported in Section 5.7).

### 4.2 StateX for SSM

The head merging method is not applicable to SSMs because there is only one key vector in each layer. For this RNN variant, we increase the key dimension by expanding the key and query projection layers. Specifically, we increase the hyperparameter $d_k$ (the original Mamba2 paper refers to this as the *state dimension*) and the parameters $\theta_k, \theta_q$ that depend on it. Since these two sets of parameters are much smaller than the other components, the increase in total parameters is less than 1% when we increase $d_k$ by $4\times$. This modification is illustrated by Figure 2 (right).

### 4.3 Parameter Initialization

After the modification, we can inherit the parameters from the pre-trained model and initialize only the added parameters (for SSMs). However, perhaps surprisingly, we find that inheriting pre-trained parameters can be detrimental to downstream performance. Thus, we present a better parameter initialization strategy.

Table 2: Accuracy on recall-intentive tasks with sequences truncated to a maximum of 2K tokens, as well as the model size and state size of each model. The best scores are bolded.

| Model | Params | Total State | SWDE | SQuAD | TQA | NQ | Drop | Avg. ↑ |
|---|---|---|---|---|---|---|---|---|
| *Linear Attention — GLA* | | | | | | | | |
| *Original Model* | 1.365B | 12.58M | 44.64 | 54.96 | 54.80 | 19.10 | 33.64 | 41.42 |
| LPT | 1.365B | 12.58M | 47.16 | 56.84 | 56.04 | 21.95 | 36.56 | 43.71 |
| StateX (ours) | 1.365B | 18.87M | 50.32 | 59.15 | 55.04 | 21.82 | 39.58 | **45.18** |
| *State Space Model — Mamba2* | | | | | | | | |
| *Original Model* | 1.343B | 24.96M | 57.43 | 59.58 | 63.27 | 5.16 | 36.22 | 44.33 |
| LPT | 1.343B | 24.96M | 54.19 | 57.81 | 63.51 | 36.87 | 35.46 | 49.56 |
| StateX (ours) | 1.350B | 37.44M | 56.17 | 57.91 | 63.68 | 36.43 | 36.37 | **50.11** |
| *Sparse Model – MoM* | | | | | | | | |
| *MoM (Du et al., 2025)* | 1.552B | 31.45M | 34.4 | 49.6 | 50.1 | 16.0 | 33.9 | 36.8 |

Table 3: In-context learning performance of GLA and Mamba2 variants, evaluated on 12 downstream classification tasks. Higher is better.

| GLA | 8-shot ↑ | 16-shot ↑ | 24-shot ↑ | Mamba2 | 8-shot ↑ | 16-shot ↑ | 24-shot ↑ |
|---|---|---|---|---|---|---|---|
| *Original* | 48.98 | 47.91 | 48.50 | *Original* | 51.40 | 54.34 | 51.60 |
| LPT | 47.33 | 49.70 | 48.45 | LPT | **47.72** | 49.79 | 52.49 |
| StateX (ours) | **48.15** | **52.42** | **51.95** | StateX (ours) | 47.68 | **52.34** | **53.03** |
| *Sparse Model – MoM (Du et al., 2025)* | | | | | | | |
| MoM | 42.6 | 42.2 | 42.9 | MoM | 42.6 | 42.2 | 42.9 |

We assume that world knowledge is usually stored in FFN blocks and the embedding table, and these parameters take longer to learn than the token-mixing parameters (GLA and SSM blocks). Thus, we reinitialize parameters that are responsible for token-mixing while other components inherit from the pre-trained checkpoint. An ablation study on initialization strategies is provided in Section 5.4.

**GLA Initialization** GLA models consist of interleaving layers of GLA blocks and FFN blocks. After state expansion, we reinitialize all parameters associated with the GLA blocks, while FFN blocks and the embedding table inherit the pre-trained parameters.

**SSM Initialization** Mamba2 merges FFN blocks and the SSM mechanism into one unified layer. Motivated by the SSM literature, we only reinitialize the parameters of the SSM mechanism, which are $A_h, \theta_k, \theta_q, \theta_{\Delta,h}$, while other modules inherit the pre-trained parameters. Further implementation details can be found in Appendix A.4.

## 4.4 HOW MANY LAYERS TO EXPAND?

Modifying all layers may result in a too disruptive change, making it harder for the modified model to recover from this change through post-training. Existing works have shown that not all layers are responsible for recalling information (Bick et al., 2025). Thus, we hypothesize that only a subset of layers can benefit from a larger state. Concretely, we adopt a uniform expansion strategy by expanding one layer every $\lfloor L/m \rfloor$ layers (where $L$ is the total number of layers), starting from the first layer, so that exactly $m$ layers are expanded. For both GLA and Mamba2, we use $m = 4$ by default. In Section 5.5, we empirically ablate the influence of the number of expanded layers.

Table 4: Performance on language modeling and zero-shot common-sense reasoning.

| Model | PIQA acc ↑ | Hella. acc ↑ | Wino. acc ↑ | ARC-e acc ↑ | ARC-c acc ↑ | SIQA acc ↑ | Avg. ↑ |
|---|---|---|---|---|---|---|---|
| *Linear Attention — GLA* | | | | | | | |
| *Original Model* | 69.70 | 38.97 | 53.35 | 55.13 | 23.38 | 39.92 | 46.74 |
| LPT | 69.64 | 38.21 | 54.78 | 54.59 | 22.70 | 39.61 | 46.58 |
| StateX (ours) | 69.75 | 37.16 | 54.93 | 53.91 | 22.53 | 39.97 | 46.37 |
| *State Space Model — Mamba2* | | | | | | | |
| *Original Model* | 73.29 | 45.89 | 60.85 | 64.31 | 30.12 | 43.14 | 52.93 |
| LPT | 73.07 | 45.48 | 59.67 | 64.31 | 29.10 | 41.10 | 52.12 |
| StateX (ours) | 73.67 | 45.09 | 59.98 | 64.02 | 29.61 | 41.61 | 52.33 |
| *Sparse Model – MoM* | | | | | | | |
| *MoM (Du et al., 2025)* | 63.3 | 30.4 | 50.8 | 45.2 | 18.8 | 37.4 | 41.0 |

## 5 EXPERIMENTS

We first describe the details of the experiments (Section 5.1). Then, we present the main results of our method (Section 5.2) as well as improvement on long-context retrieval tasks (Section 5.3). Finally, we provide ablation studies involving the choices of parameter initialization (Section 5.4), the number of expanded layers (Section 5.5), multi-head mechanism in GLA (Section 5.7). We also report the training loss in Section 5.6.

### 5.1 EXPERIMENTAL DETAILS

**Models**   We apply StateX to the official 1.3B checkpoints of GLA and Mamba2. In StateX for Mamba2, we increase the $d_k$ hyperparameter from 128 to 512. For GLA, the pre-trained 1.3B checkpoint has four heads, so StateX versions of the expanded layers have $4\times$ larger states.

**Data**   All models are trained on SlimPajama (Soboleva et al., 2023), a widely-used, high-quality, and deduplicated corpus with 627B tokens extracted from the Internet. We concatenate documents with a special token as the delimiter. Then, these concatenations are split into chunks of the specified training context length.

**Training Configuration**   The training follows common practices in context length extension by post-training as closely as possible. Concretely, we use the cosine learning rate scheduler, with a maximum learning rate of 3e-4, and a warmup phase of 5% of the total training steps. To better evaluate the ability to recall information from long contexts, we use a 64K context length. The training spans a total of 10B tokens, with a batch size of 0.5M tokens.

**Evaluation**   We evaluate the models' context utilization abilities with recall-intensive tasks and in-context learning (ICL). The recall-intensive tasks involves 5 popular document question-answering tasks. To assess ICL, we adopt a suite of 7 classification and 5 multiple-choice tasks selected from Min et al. (2022), a study that systematically evaluates ICL capabilities. Models are evaluated with accuracy across varying number of demonstrations, and ICL performance is summarized by the mean accuracy averaged over all tasks. Furthermore, we measure the general language processing abilities with 6 popular multiple-choice common-sense reasoning tasks.

More details are given in Appendix B.1.

**Baselines**   We mainly compare StateX against vanilla RNNs and the ordinary LPT versions. The LPT models undergo the same post-training process, but without any architectural modifications, so their state sizes remain unchanged.

Figure 3: Performance on retrieving specific information (i.e., a needle) from synthetically generated long documents up to 64K tokens.

| Model | 4K | 8K | 16K | 32K | 64K |
|---|---|---|---|---|---|
| *GLA — Passkey Retrieval* | | | | | |
| *Original* | 0.25 | 0.01 | 0.00 | 0.00 | 0.00 |
| LPT | 0.74 | 0.41 | 0.13 | 0.01 | **0.01** |
| StateX (ours) | **0.93** | **0.77** | **0.34** | **0.06** | **0.01** |
| *Mamba2 — NIAH-Single-2* | | | | | |
| *Original* | 0.05 | 0.00 | 0.00 | 0.00 | 0.00 |
| LPT | 0.83 | 0.43 | 0.30 | **0.09** | **0.01** |
| StateX (ours) | **0.94** | **0.61** | **0.32** | **0.09** | 0.00 |

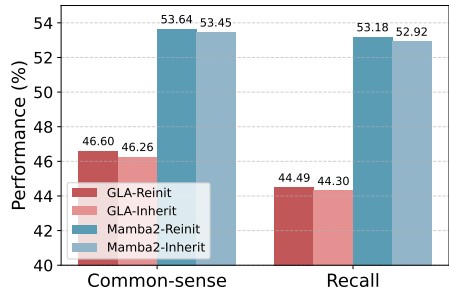

Figure 4: Model performance of reinitialization and parameter inheritance.

## 5.2 MAIN RESULTS

**Recall Abilities**  Table 2 presents scores on recall-intensive tasks for the original model (Vanilla), the model using the standard long-context post-training (LPT), and the model enhanced with StateX. The columns "Params" and "Total State" report the number of model parameters and state parameters for each model, respectively. StateX increases the total state sizes by roughly 50%. The main takeaway is that StateX models achieve the highest average performance, underscoring the advantage of larger states.

**In-Context Learning**  Table 3 shows the in-context learning performance of various RNN variants, and StateX variants exhibits significantly greater in-context learning abilities.

**Common-Sense Reasoning**  Table 4 shows that StateX models' performance on common-sense reasoning is comparable to the vanilla model, implying that pre-training knowledge remains largely unaffected by the architectural change.

## 5.3 IMPROVEMENT ON LONG-CONTEXT RETRIEVAL

The recall-intentive tasks we used in Section 5.2 contain mostly sequences with fewer than 4K tokens. To evaluate the models' abilities to retrieve information from longer contexts, we use the popular NIAH task (Hsieh et al., 2024). Due to differences in the recall abilities between the GLA and Mamba2, we evaluate them using NIAH tasks of varying difficulty to avoid score saturation and preserve discriminative resolution. For the GLA model, we employed the simpler passkey retrieval task from ∞Bench (Zhang et al., 2024), which involves retrieving a single 5-digit passkey from long documents consisting of repeated text. For Mamba2, we use the more challenging NIAH-Single-2 task from RULER (Hsieh et al., 2024), where a 7-digit passkey is embedded in a semantically meaningful, non-repetitive distractor content. More details can be found in Appendix B.3.

**Results**  Table 3 reports the models' performances in NIAH. It shows that, by unlocking a larger state size, StateX significantly improves the model's recall performance in long contexts.

## 5.4 COMPARISON BETWEEN REINITIALIZATION AND PARAMETER INHERITANCE

Although it may seem natural to inherit pre-trained parameters, our experiments show that reinitializing the modified parameters yields better performance. For Mamba2, whose state expansion process introduces new parameters, we initialize the new parameters with zeros.

As illustrated in Figure 4, the model with reinitialized parameters (Reinit) consistently outperforms the one that inherits parameters (Inherit) on both common-sense reasoning and recall tasks. We hypothesize that the performance gap arises because the inherited parameters have already converged, making it difficult to effectively utilize the newly introduced channels (indicated in red in Figure 2) via post-training.

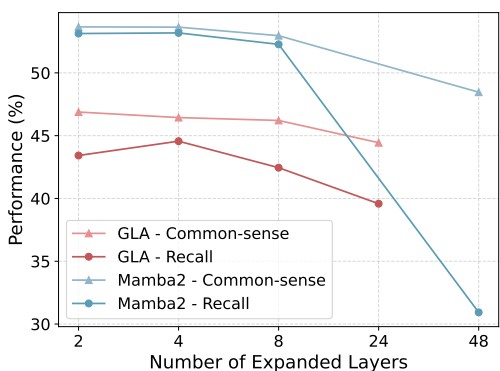 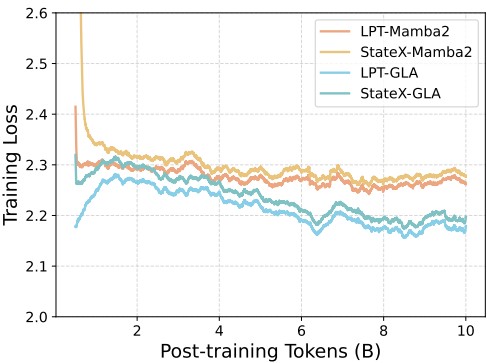

Figure 5: Model performance under varying numbers of expanded layers. Mamba2 has twice as many layers as GLA because it does not have FFN layers.

Figure 6: Post-training loss (on SlimPajama) of vanilla models and expanded models. GLA has lower loss as it is pre-trained on SlimPajama while Mamba2 is pre-trained on Pile.

## 5.5 BEST PROPORTION OF EXPANDED LAYERS

As mentioned in Section 4.4, it is important to balance the number of expanded layers. To investigate this trade-off, we conducted an ablation study by varying the number of expanded layers. The results, shown in Figure 5, indicate that both the GLA and Mamba2 models achieve optimal average performance when four layers are expanded (out of 24 layers and 48 layers, respectively). When too many layers are modified, the reinitialized parameters fail to converge effectively under limited post-training, leading to a sharp drop in overall performance.

## 5.6 TRAINING LOSS

We also tracked the training loss curves of models trained with standard LPT and with StateX. Figure 6 shows the loss curves for both GLA and Mamba2. The former has generally lower loss because it was pre-trained on SlimPajama, while Mamba2 was not. Notably, the StateX models have a higher initial training loss due to the architectural change, but quickly close the gap. Interestingly, although their final training loss is slightly higher than the LPT counterparts, they achieve better performance on downstream tasks.

## 5.7 THE OPTIMALITY OF SINGLE-HEAD GLA

As mentioned in Section 4.1, the multi-head mechanism in GLA significantly reduces the size of the recurrent state, which in turn leads to a degradation in model performance. This section presents an ablation study on the number of heads for GLA models trained from scratch.

We conducted experiments on GLA models with 340M parameters, trained on 20B tokens from the SlimPajama dataset (Soboleva et al., 2023). More experimental details are described in Section B.4. Table 5 reports the performance

Table 5: Common-sense reasoning (CSR), recall, and training loss of GLA-340M models with different numbers of heads. Single-head GLA outperforms other configurations due to larger states.

| Head number | CSR ↑ | Recall ↑ | Tr. Loss ↓ |
|---|---|---|---|
| 1 | **42.715** | **25.992** | **2.722** |
| 4 | 42.029 | 24.012 | 2.762 |
| 8 | 42.401 | 21.780 | 2.798 |
| 16 | 41.527 | 15.395 | 2.883 |

of these models on a range of common tasks. As shown, the single-head model achieves higher average scores on the benchmark tasks and converges to a lower final training loss. Given the same number of parameters and other configurations, using fewer heads allows for a larger state size, which in turn leads to improved performance in common-sense reasoning, recall, and training loss.

## 5.8 EFFICIENCY ANALYSIS OF STATEX

Although there is expansion of states in StateX, these models still have a high efficiency compared to their vanilla versions. We have measured the throughput of the vanilla GLA and Mamba2, their StateX versions, and MoM (Du et al., 2025), in training, prefilling (which is correlated with latency), and decoding.

The RNN component of each model is implemented with kernels from the widely-used flash-linear-attention GitHub repository (4K stars). Inference throughput measurements are performed on one NVIDIA A800-SXM4-80GB GPU, and training throughput is measured on a machine equipped with eight NVIDIA A800-SXM4-80GB GPU. The training framework is implemented with the popular HuggingFace Accelerate framework with data parallelism (which is a common approach for single-machine, multi-GPU training).

Table 6: Prefilling throughput (tokens/s) across different context lengths.

| Context Length | 1K | 4K | 16K | 64K | Avg. ↑ |
|---|---|---|---|---|---|
| *Linear Attention — GLA* | | | | | |
| Vanilla GLA | 72.5K | 72.6K | 72.2K | 70.4K | 71.9K |
| StateX GLA (ours) | 70.0K | 70.1K | 69.5K | 68.0K | 69.4K |
| *State Space Model — Mamba* | | | | | |
| Vanilla Mamba | 44.0K | 44.1K | 44.2K | 50.7K | 45.7K |
| StateX Mamba (ours) | 45.1K | 45.3K | 42.6K | 52.2K | 46.3K |
| *Sparse Model — MoM* | | | | | |
| MoM | 20.3K | 28.1K | 29.5K | 31.4K | 27.3K |

Table 7: Decoding throughput (tokens/s) across different batch sizes.

| Model | BSZ=64 | BSZ=128 | BSZ=256 | BSZ=512 | Average ↑ |
|---|---|---|---|---|---|
| *Linear Attention — GLA* | | | | | |
| Vanilla GLA | 3548.1 | 3814.6 | 9594.8 | 10225.9 | 6795.9 |
| StateX-GLA (ours) | 3769.6 | 6371.1 | 7082.5 | 7394.9 | 6154.5 |
| *State Space Model — Mamba* | | | | | |
| Vanilla Mamba | 2275.5 | 3033.7 | 4754.9 | 5730.0 | 3948.5 |
| StateX-Mamba (ours) | 2173.5 | 3120.6 | 4350.8 | 4836.8 | 3620.4 |
| *Sparse Model — MoM* | | | | | |
| MoM | 46.3 | 47.5 | 48.2 | *OOM* | 47.3 |

It can be shown that StateX versions of GLA and Mamba2 almost as fast as the original models in prefilling and training, and slightly slower in decoding. Compared to MoM, StateX is:

- roughly 1.9x to 2.2x faster in training (Section B.5);
- roughly 1.7x to 2.5x faster in prefilling (Table 6);
- roughly up to 147x faster (using batch size of 256) during decoding, while MoM gets out of CUDA memory in the batch size of 512 (Table 7).

## 6 CONCLUSIONS

We have proposed StateX, a novel method for enhancing the recall abilities of two popular RNN variants by expanding the state sizes of pre-trained RNNs through post-training. Compared to training RNNs with larger state sizes from scratch, our method is much faster to train and can be seamlessly applied to existing pre-trained models of said RNN variants. StateX is valuable for closing the gap in the recall abilities of RNNs and Transformers, especially in long-context scenarios. This work represents an important step toward RNNs as an efficient alternative to attention-based architectures.

REPRODUCIBILITY STATEMENT

We have included the code for reproducing our results as supplementary materials. We will release the model checkpoints after the anonymous period.

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

Table 8: Overview of GLA and Mamba2, two popular RNNs with matrix-valued recurrent states. $H, P, N, d_k, d_v$ are hyperparameters of the architectures. $E$ is the expansion ratio of StateX for SSMs, which is set to 4, as mentioned in Section 4.2

| Model | Update rule | Query rule | State size | StateX state size |
|---|---|---|---|---|
| GLA | $\mathbf{S}_{t-1,h}\mathrm{diag}(\alpha_{t,h}) + \mathbf{k}_{t,h}^T\mathbf{v}_{t,h}$ | $\mathbf{q}_{t,h}\mathbf{S}_{t,h}$ | $Hd_kd_v$ | $H^2d_kd_v$ |
| Mamba2 | $\mathbf{S}_{t-1,h}\alpha_{t,h} + \Delta_{t,h}\mathbf{k}_t^T\mathbf{v}_{t,h}$ | $\mathbf{q}_t\mathbf{S}_{t,h} + D_h\mathbf{v}_{t,h}$ | $Hd_kd_v$ | $Hd_vd_kE$ |

# A  FORMULATION OF GATED LINEAR ATTENTION AND MAMBA2

For completeness, we provide the complete formulation of GLA and Mamba2 in this section. These models are trained on the next-token prediction task, which means that their input is a sequence of token IDs and their output is a sequence of probability distributions over the vocabulary $\{1, \cdots, V\}$, where $V$ is the vocabulary size.

At the beginning, each token ID is converted to a $d$-dimensional token embedding by looking up an embedding table (also called the *input embeddings*) before passing to the backbone network. Let $T$ denote the sequence length. This creates a sequence of $T$ embeddings $\mathbf{X}^{(0)} \in \mathbb{R}^{T \times d}$. On the output side, the output embeddings at each position $t \in \{1, \cdots, T\}$ are converted to a probability distribution over the vocabulary via a linear layer called the *language modeling head*.

In the following discussion, we denote the input and output sequences of representations for the $l$-th layer as:

$$\mathbf{X}^{(l)} = \begin{bmatrix} \mathbf{x}_1^{(l)} \\ \vdots \\ \mathbf{x}_T^{(l)} \end{bmatrix}, \mathbf{Y}^{(l)} = \begin{bmatrix} \mathbf{y}_1^{(l)} \\ \vdots \\ \mathbf{y}_T^{(l)} \end{bmatrix} \tag{4}$$

where $T$ is the sequence length, and $\mathbf{x}_t^{(l)}, \mathbf{y}_t^{(l)} \in \mathbb{R}^{1 \times d}$ are the input and output representations at time step $t$. Since the input of each layer is the output of the previous layer, we have $\mathbf{X}^{(l)} = \mathbf{Y}^{(l-1)}$.

## A.1  GATED LINEAR ATTENTION

The entire model of GLA consists of interleaving GLA blocks and FFN blocks.

$$\mathbf{Y}'^{(l)} = \mathrm{GLA}^{(l)}\left(\mathbf{X}^{(l-1)}\right) + \mathbf{X}^{(l-1)}$$

$$\mathbf{Y}^{(l)} = \mathrm{FFN}^{(l)}\left(\mathbf{Y}'^{(l)}\right) + \mathbf{Y}'^{(l)} \tag{5}$$

Each GLA block consists of multiple heads that are computed in parallel, and the block's output is the sum of the head outputs. This can be formulated as (omitting the layer index for simplicity):

$$\mathbf{y}_t = \sum_{h=1}^{H} \mathrm{GLA}_h(\mathbf{x}_t) \tag{6}$$

Each head in GLA can be formulated as:

$$\begin{aligned} \Box_{t,h} &= \mathbf{x}_t\mathbf{W}_\Box, \quad \Box \in \{\mathbf{q},\mathbf{k},\mathbf{v},\boldsymbol{\alpha}\}, \\ \mathbf{S}_{t,h} &= \mathrm{diag}(\boldsymbol{\alpha}_{t,h})S_{t-1,h} + \mathbf{k}_{t,h}^\top\mathbf{v}_{t,h}, \\ \mathbf{o}_{t,h} &= \mathrm{LN}(\mathbf{q}_{t,h}S_{t,h}), \\ \mathbf{r}_t &= \mathrm{SILU}(\mathbf{x}_t W_r + b_r), \\ \mathrm{GLA}_h(\mathbf{x}_t) &= (\mathbf{r}_t \odot \mathbf{o}_{t,h})\mathbf{W}_o. \end{aligned} \tag{7}$$

## A.2  MAMBA2

Mamba2 does not have FFNs and consists only of a stack of Mamba2 blocks:

$$\mathbf{Y}^{(l)} = \mathrm{Mamba2}^{(l)}\left(\mathbf{X}^{(l)}\right) + \mathbf{X}^{(l)} \tag{8}$$

Mamba2 also employs a multi-head mechanism where the layer output is the sum of the head outputs (omitting the layer index for simplicity):

$$\text{Mamba2}(\mathbf{x}_t) = \sum_{h=1}^{H} \text{Mamba2}_h(\mathbf{x}_t) \tag{9}$$

where $H$ is the number of heads, and $h$ is the head index. Each Mamba2 head can be formulated as:

$$
\begin{aligned}
\mathbf{v}_{t,h} &= f_v(\mathbf{x}_t, \theta_{v,h}) \in \mathbb{R}^{d_v} \\
\mathbf{k}_t &= f_k(\mathbf{x}_t, \theta_k) \in \mathbb{R}^{d_k} \\
\mathbf{q}_t &= f_q(\mathbf{x}_t, \theta_q) \in \mathbb{R}^{d_k} \\
\Delta_{t,h} &= \text{SILU}(\mathbf{x}_t \mathbf{W}_{\Delta,h} + \mathbf{b}_{\Delta,h}) \in \mathbb{R} \\
\alpha_{t,h} &= \exp(-\Delta_t A_h) \in \mathbb{R} \\
\mathbf{S}_{t,h} &= \mathbf{S}_{t-1,h} \alpha_{t,h} + \Delta_{t,h} \mathbf{k}_t^\top \mathbf{v}_{t,h} \in \mathbb{R}^{d_k \times d_v} \\
\mathbf{o}_{t,h} &= \mathbf{q}_t \mathbf{S}_{t,h} + D_h \mathbf{v}_{t,h} \in \mathbb{R}^{d_v} \\
\mathbf{z}_{t,h} &= \text{SILU}(\mathbf{x}_t \mathbf{W}_{z,h}) \in \mathbb{R}^{d_v} \\
\mathbf{y}_{t,h} &= \text{Norm}(\mathbf{o}_{t,h} \odot \mathbf{z}_{t,h}) \mathbf{W}_{o,h} \in \mathbb{R}^{d}
\end{aligned}
\tag{10}
$$

### A.3 UPDATE RULE AND QUERY RULE

Central to recurrent architectures are the update rule and query rule (described in Section 3.1), which dictate how the architecture models inter-token dependencies. Table 8 shows the update rule and query rule of GLA and Mamba2.

### A.4 DETAILS OF PARAMETER REINITIALIZATION

In the case of GLA, we reinitialize all parameters within the GLA block, including its normalization layer. For Mamba, we reinitialize all parameters of $A_h, \theta_k, \theta_q$. And $\theta_{\Delta,h}$ is reinitialized specifically by resetting its internal dt_bias component.

## B EXPERIMENT DETAILS

### B.1 EVALUATION

We configure the evaluation tasks using the lm-evaluation-harness framework Gao et al. (2024). A set of widely adopted benchmark tasks is selected to assess the models' capabilities in common-sense reasoning and information recall. For the common-sense and recall tasks, we adopt *accuracy* (not *normalized accuracy*) and *contains* as the respective evaluation metrics. *Accuracy* directly reflects the correctness of the common-sense task results, while *contains* measures the proportion of recall task outputs that include the passkey. Notably, for tasks related to recall ability, we adopt the Just Read Twice prompt from Arora et al. (2024b), which is also used in Yang et al. (2024) and Yang et al. (2025), given that all models under evaluation are based on recurrent architectures.

### B.2 IN-CONTEXT LEARNING EVALUATION

For the in-context learning (ICL) evaluation, we follow the setup introduced by Min et al. (2022), which systematically benchmarks ICL capabilities across classification and multiple-choice tasks. Our evaluation adopts the same protocol, but we evaluate also evaluate with different number of in-context demonstrations for comprehensiveness.

The tasks that were used for evaluation are:

- `commonsense_qa`
- `ai2_arc`
- `superglue-copa`

- `superglue-cb`
- `glue-mrpc`
- `glue-sst2`
- `glue-qqp`
- `glue-cola`
- `superglue-rte`
- `superglue-wic`
- `codah`
- `dream`

### B.3 NEEDLE-IN-A-HAYSTACK TASKS

As mentioned in the previous section, we design two passkey retrieval tasks with varying levels of difficulty. The specific noise configurations and prompt templates used in each task are detailed in Table 9. We use 5-digit passkeys in Passkey Retrieval and 7-digit passkeys in NIAH-Single-2. For each unique test length, the task will be tested on 256 randomly generated examples to ensure the consistency of the results.

Table 9: The prompt templates of the NIAH tasks used to evaluate the models in retrieving information from long contexts.

| **Passkey Retrieval (∞Bench)** | **Task Template:** |
| --- | --- |
| | The grass is green. The sky is blue. The sun is yellow. Here we go. There and back again. |
| | ...... |
| | The pass key is {number}. Remember it. {number} is the pass key. |
| | ...... |
| | The grass is green. The sky is blue. The sun is yellow. Here we go. There and back again. |
| | |
| | **Task Answer Prefix:** |
| | What is the pass key? The pass key is |
| **NIAH-Single-2 (RULER)** | **Task Template:** |
| | Some special magic numbers are hidden within the following text. Make sure to memorize it. I will quiz you about the numbers afterwards. |
| | Paul Graham Essays. |
| | ...... One of the special magic numbers for {word} is: {number}. ...... What is the special magic number for {word} mentioned in the provided text? |
| | |
| | **Task Answer Prefix:** |
| | The special magic number for {word} mentioned in the provided text is |

### B.4 MORE DETAILS: ABLATION STUDY ON THE NUMBER OF GLA HEADS

The training procedure for these models follows common language model pre-training practices as closely as possible. The model is trained on 20B tokens from SlimPajama, with a 0.5M tokens per batch, and a sequence length of 4k. We employ a cosine learning rate scheduler with an initial learning rate of 3e-4 and no specified minimum learning rate. All models consist of 340 million parameters and comprise 24 layers, each with an identical hidden state dimension. The only architectural difference lies in the number of attention heads: the single-head model uses one head with

a dimensionality of 512, while the four-head model uses four heads, each with a dimensionality of 128, and so on, following the same principle.

## B.5 EFFICIENCY DETAILS

Table 10: Training throughput of vanilla models, StateX models and MoM. The StateX models have a close throughput to vanilla ones, while they are roughly 2x faster than MoM.

| Model | Vanilla GLA | StateX-GLA | Vanilla Mamba | StateX-Mamba | MoM |
|---|---|---|---|---|---|
| Throughput (tokens/s) | 129.1K | 122.1K | 108.5K | 104.3K | 55.9K |

## C THE USE OF LARGE LANGUAGE MODELS

Large language models (LLMs) were used to quality-check the final draft, but we never explicitly instruct LLMs to write any parts of this paper.

