# OpenReview forum: "StateX: Enhancing RNN Recall via Post-training State Expansion"
_ICLR.cc/2026/Conference — ICLR 2026 Conference Withdrawn Submission_

### Official Review · Reviewer_97sA · 2025-10-31

**Soundness:** 2
**Presentation:** 3
**Contribution:** 2
**Rating:** 4
**Confidence:** 5

**Summary:**

This paper explores a method for expanding the state size of linear attention mechanisms, which is characterized as follows:
- (1) Keeping the state size unchanged during the pre-training phase and just expanding it before post-training, which partially reduces training costs compared to expanding the state size throughout the entire process.
- (2) The approach belongs to a common dense state expansion method: For GLA, multiple attention heads are merged into one; For Mamba2, the state dimension and the corresponding projection parameters are directly expanded.
- (3) Two approaches for handling parameters after state expansion, namely parameter inheritance and re-parameterization, are investigated.
- (4) The authors have also explored how to selectively expand the size of a portion of the layers.

In terms of experimental setup, this paper mainly conducts the following studies:
- (1) Post-training is performed on both standard GLA and GLA after state expansion, comparing their language modeling, zero-shot common-sense reasoning, in-context learning, and retrieval abilities.
- (2) For the two methods of parameter inheritance and re-parameterization after state expansion, their retrieval capabilities are compared.
- (3) Ablation experiments are conducted on the number of attention heads and the number of layers where state expansion is applied. Compared with models without state expansion, the post-trained models with state expansion achieve varying degrees of performance improvement on various benchmarks.

**Strengths:**

(1) Compared with expanding the state size throughout the entire training process, selectively expanding the state size of certain layers during the post-training stage can effectively reduce the overall state size and the computational cost during training.

(2) The re-parameterization technique after state expansion have been shown to be effective to a certain extent.

**Weaknesses:**

The paper lacks comparisons with other state expansion approaches. Previous works (MoM[1], SSE[2]) have already explored sparse expansion methods (e.g., increasing hidden state size by 16 times but only increasing activated state size by 2–4 times), and have investigated and interpreted the functional roles of various sparsely activated hidden state units. Comparisons with these baselines would make the proposed method more convincing.

**References:**
[1] https://arxiv.org/abs/2502.13685
[2] https://arxiv.org/abs/2507.16577

**Questions:**

See the weaknesses part.

---

> ### Author Response · Authors · 2025-11-25
> **Response to Reviewer 97sA | Part 1 of 2**
>
> Dear reviewer,
>
> We greatly appreciate your time and effort. To address the weakness that you have raised, we have conducted additional comparisons with MoM [1]. We do not compare against SSE [2] because their code has not been open sourced.
>
> **Your potential misunderstanding about MoM**
>
> We would like to emphasize that MoM is pre-trained from scratch (with no capability to inherit pre-trained weights), while StateX is applied to pre-trained checkpoints. It is common to use trillions of tokens for training language models from scratch, while StateX's training process only takes 10B tokens. Thus, these two models are not directly comparable.
>
> ## Comparison Against MoM
>
> Although MoM is not a post-training method, we decided to compare against them for completeness.
>
> To ensure fair comparison, we train an MoM model from scratch using the same data we used for StateX. The training hyperparameters are also identical. Then, we compare StateX with it on common-sense reasoning, recall-intensive tasks, and in-context learning. Finally, we also measured the training and inference efficiency of StateX and MoM for comparison.
>
> ### Performance Results
>
> The results are presented in the following three tables. It is evident that MoM performs much worse than StateX in terms of common-sense reasoning, recall abilities, and in-context learning, when **trained on the same amount of data**. Also, MoM has a much lower training efficiency; its training time is twice as long as StateX's training time for the same amount of data (see below for efficiency measurements).
>
> **Common-Sense Reasoning**
>
> | **Model**           | **PIQA** | **Hella** | **Wino** | **ARC-e** | **ARC-c** | **SIQA** | **Average** |
> | :------------------ | :------- | :-------- | :------- | :-------- | :-------- | :------- | :---------- |
> | StateX-GLA (ours)   | 69.8     | 37.2      | 54.9     | 53.9      | 22.5      | 40.0     | 46.4        |
> | StateX-Mamba (ours) | 73.7     | 45.1      | 60.0     | 64.0      | 29.6      | 41.6     | 52.3        |
> | MoM                 | 63.3     | 30.4      | 50.8     | 45.2      | 18.8      | 37.4     | 41.0        |
>
> **Recall Abilities**
>
> | **Model**           | **SWDE** | **SQuAD** | **TQA** | **NQ** | **Drop** | **Average** |
> | :------------------ | :------- | :-------- | :------ | :----- | :------- | :---------- |
> | StateX-GLA (ours)   | 50.3     | 59.2      | 55.0    | 21.8   | 39.6     | 45.2        |
> | StateX-Mamba (ours) | 56.2     | 57.9      | 63.7    | 36.4   | 36.4     | 50.1        |
> | MoM                 | 34.4     | 49.6      | 50.1    | 16.0   | 33.9     | 36.8        |
>
> **In-Context Learning**
>
> | **Model**           | **8-shot** | **16-shot** | **24-shot** | **32-shot** | **Average** |
> | :------------------ | :--------- | :---------- | :---------- | :---------- | :---------- |
> | StateX-GLA (ours)   | 48.2       | 52.4        | 52.0        | 50.6        | 50.8        |
> | StateX-Mamba (ours) | 47.7       | 52.3        | 53.0        | 49.8        | 50.7        |
> | MoM                 | 42.6       | 42.2        | 42.9        | 42.3        | 42.5        |
>
> **Takeaways.**
> StateX outperforms MoM on common-sense reasoning, recall-intensive tasks, and in-context learning by a large margin, when trained on the same data.

---

> ### Author Response · Authors · 2025-11-25
> **Response to Reviewer 97sA | Part 2 of 2**
>
> ### Efficiency Results
>
> **Experimental Settings**
>
> The RNN component of each model is implemented with kernels from the widely used flash-linear-attention GitHub repository (4K stars). Inference throughput measurements are performed on one NVIDIA A800-SXM4-80GB GPU, and training throughput is measured on a machine equipped with eight NVIDIA A800-SXM4-80GB GPU. The training framework is implemented with the popular HuggingFace Accelerate framework with data parallelism (which is popular for single-machine, multi-GPU training).
>
> **Training Efficiency Results**
>
> The following table reports the training throughput for the original models, StateX models, and MoM.
>
> | **Model**           | **Vanilla GLA** | **StateX GLA** | **Vanilla Mamba** | **StateX Mamba** | **MoM** |
> | :------------------ | :-------------- | :------------- | :---------------- | :--------------- | :------ |
> | Training Throughput | 129.1K          | 122.1K         | 108.5K            | 104.3K           | 55.9K   |
>
> **Inference Efficiency Results**
>
> The following table reports the prefilling throughput for various context lengths T. For consistency and practicality, we adjust the batch size B such that B*T is constant.
>
> | **Model**               | **1K ctx. len.** | **4K ctx. len.** | **16K ctx. len.** | **64K ctx. len.** | **Average** |
> | :---------------------- | :--------------- | :--------------- | :---------------- | :---------------- | :---------- |
> | *Vanilla GLA*           | *72.5K*          | *72.6K*          | *72.2K*           | *70.4K*           | *71.9K*     |
> | **StateX-GLA (ours)**   | **70.0K**        | **70.1K**        | **69.5K**         | **68.0K**         | **69.4K**   |
> | *Vanilla Mamba*         | *44.0K*          | *44.1K*          | *44.2K*           | *50.7K*           | *45.7K*     |
> | **StateX-Mamba (ours)** | **45.1K**        | **45.3K**        | **42.6K**         | **52.2K**         | **46.3K**   |
> | MoM                     | 20.3K            | 28.1K            | 29.5K             | 31.4K             | 27.3K       |
>
> The following table reports the decoding throughput for various batch sizes. For RNN models, the decoding speed is independent of the context length.
>
> | **Model**               | **BSZ = 64** | BSZ = 128  | **BSZ = 256** | BSZ = 512                | **Average** |
> | :---------------------- | :----------- | ---------- | :------------ | ------------------------ | :---------- |
> | *Vanilla GLA*           | *3548.1*     | 3814.6     | *9594.8*      | 10225.9                  | 6795.9      |
> | **StateX-GLA (ours)**   | **3769.6**   | **6371.1** | **7082.5**    | **7394.9**               | **6154.5**  |
> | *Vanilla Mamba*         | *2275.5*     | 3033.7     | *4754.9*      | 5730.0                   | 3948.5      |
> | **StateX-Mamba (ours)** | **2173.5**   | **3120.6** | **4350.8**    | **4836.8**               | **3620.4**  |
> | MoM                     | 46.3         | 47.5       | 48.2          | ***Out of CUDA Memory*** | 47.3        |
>
> *Note: The average throughput of MoM only accounts for the BSZ of 64, 128, and 256, since MoM is out of CUDA memory (80GB) when BSZ = 512.*
>
> **Takeaway.** StateX versions of GLA and Mamba2 almost as fast as the original models in prefilling and training, and slightly slower in decoding. Compared to MoM, StateX is:
>
> - roughly 1.9x to 2.2x faster in training;
> - roughly 1.7x to 2.5x faster in prefilling;
> - roughly up to 147x faster (using batch size of 256) during decoding, while MoM gets out of CUDA memory in the batch size of 512.
>
> If you have any more questions or concerns, we are eager to engage in further discussions.
>
> Thank you in advance.
>
> **References**
>
> [1] MoM: Linear Sequence Modeling with Mixture-of-Memories.
>
> [2] Scaling Linear Attention with Sparse State Expansion

---

> ### Author Response · Authors · 2025-11-27
>
> Dear Reviewer 97sA,
>
> We genuinely appreciate your time and effort in reviewing our paper.
>
> To address your concerns, we've added a detailed response supported by extensive experiments. We would like to know if you have any other unaddressed concerns/questions and would be glad to have a further discussion. If all your concerns are resolved, we kindly hope it will be reflected with a more favorable rating on our paper.
>
> Thank you,
>
> Authors

---

### Official Review · Reviewer_oKyz · 2025-11-01

**Soundness:** 4
**Presentation:** 3
**Contribution:** 3
**Rating:** 4
**Confidence:** 5

**Summary:**

This paper introduces StateX, a novel training pipeline designed to enhance the long-context recall abilities of Recurrent Neural Networks (RNNs), specifically Gated Linear Attention (GLA) and State Space Models (SSMs). The core idea is to expand the recurrent state size of pre-trained RNNs through a post-training architectural modification step, which is performed before long-context post-training (LPT). The authors propose two state expansion methods for GLA and Mamba2, aiming to scale up state size with minimal or negligible increases in model parameters and training costs. Experiments on 1.3B parameter models demonstrate that StateX significantly improves recall and in-context learning, particularly in long-context scenarios, without compromising performance on common-sense reasoning tasks.

**Strengths:**

1. The paper introduces a novel concept of "state expansion" via post-training, which is a significant departure from previous methods that typically require training larger state RNNs from scratch. This approach promises lower training costs and easier integration into existing pre-trained models.
2. StateX directly addresses a known weakness of RNNs—their limited long-context recall capability due to fixed-size recurrent states. By efficiently expanding the state size, the method effectively improves performance on recall-intensive tasks and in-context learning.
3. The proposed methods are designed for two popular and distinct classes of RNNs: linear attention models (GLA) and state space models (Mamba2). This demonstrates the versatility and generalizability of the StateX pipeline.
4. The paper provides strong empirical evidence, with experiments on 1.3B parameter models, showing consistent improvements across various benchmarks, including recall-intensive tasks, in-context learning, and needle-in-a-haystack (NIAH) tasks.

**Weaknesses:**

1. "MoM: Linear Sequence Modeling with Mixture-of-Memories" is a good work also focus on expanding RNN memory states, but this paper did not compare with it. I recommend the authors to compare their algorithm difference and experimental performance.
2. While covering GLA and Mamba2, the paper doesn't discuss the applicability or potential challenges of StateX to other prominent RNN variants (e.g., RWKV, traditional LSTMs/GRUs). Expanding the analysis to a broader range of RNNs would strengthen the claim of a general solution.
3. While Figure 2 illustrates the concept, a more detailed explanation of the specific architectural changes and how they are implemented for both GLA and Mamba2 would be beneficial. For instance, how "merging multiple heads into one larger head" is architecturally realized in GLA could be elaborated.
4. The paper mentions that StateX is post-trained on "much less data than pre-training." While this is a strength in terms of efficiency, the exact relationship between the amount of post-training data, the magnitude of state expansion, and final performance could be further investigated.
5. The "Vanilla RNNs (small states)" are marked with 'X' for performance and '✓' for efficient training. However, the abstract and introduction claim that RNNs struggle with recall. Clarifying the baseline "Vanilla RNNs" and their typical state sizes relative to the models evaluated (e.g., Mamba2-2.8B and GLA-1.3B) would provide better context.
6. The meaning of "?" for "Novel architectures with large states" under "Easy Adoption" is a bit vague ("yet to be extensively tested at scale"). Providing more concrete reasons or examples would be helpful.
7. While the training configuration for LPT is described, more specific details about how the long-context corpus differs from the pre-training corpus, beyond just context length, could be useful.
8. While the paper mentions that "blindly increasing the state size can lead to high training and inference costs," it mainly focuses on training costs and parameters. A more explicit discussion or analysis of how StateX impacts inference costs (e.g., memory usage or latency) compared to traditional LPT or even Transformers would be valuable.

**Questions:**

see Weaknesses

---

> ### Author Response · Authors · 2025-11-25
> **Response to Reviewer oKyz| Part 1 of 2**
>
> Dear reviewer,
>
> Thank you for your detailed review. We will respond to the weaknesses you have raised one by one below.
>
> ## Weakness 1
>
> We would like to highlight that the previous state expansion works (e.g., MoM [1], LaCT [2] etc.) do not expand the states through post-training. Their models are pre-trained from scratch. Therefore, our work is the first study that expands RNN states via post-training.
>
> Nonetheless, we have conducted a comparison with MoM, which is the most relevant model with larger states. To ensure fair comparison, we train an MoM model from scratch using the same data we used for StateX. The training hyperparameters are also identical. Then, we compare StateX with it on common-sense reasoning, recall-intensive tasks, and in-context learning. Finally, we also measured the training and inference efficiency of StateX and MoM for comparison.
>
> ### Performance Results
>
> The results are presented in the following three tables. It is evident that MoM performs much worse than StateX in terms of common-sense reasoning, recall abilities, and in-context learning, when **trained on the same amount of data**. Also, MoM has a much lower training efficiency; its training time is twice as long as StateX's training time for the same amount of data (see below for throughput measurements).
>
> **Common-Sense Reasoning**
>
> | **Model**           | **PIQA** | **Hella** | **Wino** | **ARC-e** | **ARC-c** | **SIQA** | **Average** |
> | :------------------ | :------- | :-------- | :------- | :-------- | :-------- | :------- | :---------- |
> | StateX-GLA (ours)   | 69.8     | 37.2      | 54.9     | 53.9      | 22.5      | 40.0     | 46.4        |
> | StateX-Mamba (ours) | 73.7     | 45.1      | 60.0     | 64.0      | 29.6      | 41.6     | 52.3        |
> | MoM                 | 63.3     | 30.4      | 50.8     | 45.2      | 18.8      | 37.4     | 41.0        |
>
> **Recall Abilities**
>
> | **Model**           | **SWDE** | **SQuAD** | **TQA** | **NQ** | **Drop** | **Average** |
> | :------------------ | :------- | :-------- | :------ | :----- | :------- | :---------- |
> | StateX-GLA (ours)   | 50.3     | 59.2      | 55.0    | 21.8   | 39.6     | 45.2        |
> | StateX-Mamba (ours) | 56.2     | 57.9      | 63.7    | 36.4   | 36.4     | 50.1        |
> | MoM                 | 34.4     | 49.6      | 50.1    | 16.0   | 33.9     | 36.8        |
>
> **In-Context Learning**
>
> | **Model**            | **8-shot** | **16-shot** | **24-shot** | **32-shot** | **Average** |
> | :------------------- | :--------- | :---------- | :---------- | :---------- | :---------- |
> | StateX-GLA (ours)    | 48.2       | 52.4        | 52.0        | 50.6        | 50.8        |
> | StateX-Mamba (ours) | 47.7       | 52.3        | 53.0        | 49.8        | 50.7        |
> | MoM                  | 42.6       | 42.2        | 42.9        | 42.3        | 42.5        |
>
> **Takeaways.**
> StateX outperforms MoM on common-sense reasoning, recall-intensive tasks, and in-context learning by a large margin, when trained on the same data.
>
> ### Efficiency Results
>
> StateX versions of RNN models also outperform MoM in terms of training and inference efficiency by a large margin. To conserve space, these details have been consolidated and will be provided in our full response to *Weakness 8*.
>
> ## Weakness 2
>
> RWKV-5 and RWKV-6 are instanced of (gated) linear attention, thus our StateX for linear attention is directly applicable to these models. For RWKV-7, they are in the same class as (gated) DeltaNet [1]. We will update the paper to include a discussion on the applicability of StateX to this class of RNN models. In brief, StateX for DeltaNet works similarly to StateX for linear attention, which involves merging multiple smaller heads into one larger head.
>
> ## Weakness 3
>
> We will update our paper to include a rigorous formulation of our method for both GLA and Mamba2. Thank you for the suggestion.
>
> ## Weakness 4
>
> Our work is the first study that explores RNN state expansion via post-training. While it is interesting to include ablation studies regarding the expansion magnitude and post-training data size, such studies are outside the scope of our work.

---

> ### Author Response · Authors · 2025-11-25
> **Response to Reviewer oKyz| Part 2 of 2**
>
> ## Weakness 5
>
> We urge the reviewer to re-examine Table 1. Having an "X" mark for performance is consistent with the claim that RNNs struggle with recall. "Vanilla RNNs" refers to the RNN models described in the first part of the Section 2 (starting from Line 98). We have also described the state sizes for the models investigated in the paper in Line 43 and Table 1.
>
> ## Weakness 6
>
> We would like to emphasize that you have misread Table 1. For "Novel architectures with large states", the mark for "Easy Adoption" is "X", which indicates that it is not easy to adopt these architectures into existing implementations.
>
> ## Weakness 7
>
> The data used to train the LPT models is exactly the same as the data used for StateX, and we have described the data in Section 5.1, starting from Line 345. If necessary, we respectfully request you to clarify what details you think are missing in our descriptions.
>
> ## Weakness 8
>
> We have measured the throughput of the original GLA and Mamba2, their StateX versions, and MoM [1], in training, prefilling (which is also correlated with latency), and decoding.
>
> **Experimental Settings**
>
> The RNN component of each model is implemented with kernels from the widely used flash-linear-attention GitHub repository (4K stars). Inference throughput measurements are performed on one NVIDIA A800-SXM4-80GB GPU, and training throughput is measured on a machine equipped with eight NVIDIA A800-SXM4-80GB GPU. The training framework is implemented with the popular HuggingFace Accelerate framework with data parallelism (which is popular for single-machine, multi-GPU training).
>
> **Training Efficiency Results**
>
> The following table reports the training throughput (tokens per second) for
>
> | **Model**           | **Vanilla GLA** | **StateX-GLA** | **Vanilla Mamba** | **StateX-Mamba** | **MoM** |
> | :------------------ | :-------------- | :------------- | :---------------- | :--------------- | :------ |
> | Training Throughput | 129.1K          | 122.1K         | 108.5K            | 104.3K           | 55.9K   |
>
> **Inference Efficiency Results**
>
> The following table reports the prefilling throughput (tokens per second) for various context lengths $T$. For consistency and practicality, we adjust the batch size $B$ such that $B\times T$ is constant.
>
> | **Model**               | **1K ctx. len.** | **4K ctx. len.** | **16K ctx. len.** | **64K ctx. len.** | **Average** |
> | :---------------------- | :--------------- | :--------------- | :---------------- | :---------------- | :---------- |
> | *Vanilla GLA*           | *72.5K*          | *72.6K*          | *72.2K*           | *70.4K*           | *71.9K*     |
> | **StateX-GLA (ours)**   | **70.0K**        | **70.1K**        | **69.5K**         | **68.0K**         | **69.4K**   |
> | *Vanilla Mamba*         | *44.0K*          | *44.1K*          | *44.2K*           | *50.7K*           | *45.7K*     |
> | **StateX-Mamba (ours)** | **45.1K**        | **45.3K**        | **42.6K**         | **52.2K**         | **46.3K**   |
> | MoM                     | 20.3K            | 28.1K            | 29.5K             | 31.4K             | 27.3K       |
>
> The following table reports the decoding throughput for various batch sizes. For RNN models, the decoding speed is independent of the context length.
>
> | **Model**               | **BSZ = 64** | BSZ = 128  | **BSZ = 256** | BSZ = 512                | **Average** |
> | :---------------------- | :----------- | ---------- | :------------ | ------------------------ | :---------- |
> | *Vanilla GLA*           | *3548.1*     | 3814.6     | *9594.8*      | 10225.9                  | 6795.9      |
> | **StateX-GLA (ours)**   | **3769.6**   | **6371.1** | **7082.5**    | **7394.9**               | **6154.5**  |
> | *Vanilla Mamba*         | *2275.5*     | 3033.7     | *4754.9*      | 5730.0                   | 3948.5      |
> | **StateX-Mamba (ours)** | **2173.5**   | **3120.6** | **4350.8**    | **4836.8**               | **3620.4**  |
> | MoM                     | 46.3         | 47.5       | 48.2          | ***Out of CUDA Memory*** | 47.3        |
>
> *Note: The average throughput of MoM only accounts for the BSZ of 64, 128, and 256, since MoM is out of CUDA memory (80GB) when BSZ = 512.*
>
> **Takeaway.** StateX versions of GLA and Mamba2 almost as fast as the original models in prefilling and training, and slightly slower in decoding. Compared to MoM, StateX is:
>
> - roughly 1.9x to 2.2x faster in training;
> - roughly 1.7x to 2.5x faster in prefilling;
> - roughly up to 147x faster (using batch size of 256) during decoding, while MoM gets out of CUDA memory in the batch size of 512.
>
> We will add the above results to the revised version of the paper for reference.
>
> If you have any unaddressed concerns, we are eager to engage in further discussions.
>
> **References**
>
> [1] Parallelizing Linear Transformers with the Delta Rule over Sequence Length.

---

> ### Author Response · Authors · 2025-11-27
>
> Dear Reviewer oKyz,
>
> We genuinely appreciate your time and effort in reviewing our paper.
>
> To address your concerns, we've added a detailed response supported by extensive experiments. We would like to know if you have any other unaddressed concerns/questions and would be glad to have a further discussion. If all your concerns are resolved, we kindly hope it will be reflected with a more favorable rating on our paper.
>
> Thank you,
>
> Authors

---

### Official Review · Reviewer_ambX · 2025-11-03

**Soundness:** 3
**Presentation:** 2
**Contribution:** 2
**Rating:** 2
**Confidence:** 3

**Summary:**

The paper proposes **StateX**, a post-training “state expansion” recipe for RNN LLMs (GLA and Mamba2). For GLA, it **merges all heads into one larger head** (increasing effective state without adding parameters); for Mamba2, it **expands the key/query state dimension** with <1% parameter growth. Authors reinitialize token-mixing parameters while keeping embeddings/FFNs, and expand only a subset of layers. On 1.3B-parameter checkpoints with 10B post-training tokens and 64K context, StateX improves recall/ICL metrics while roughly preserving common-sense reasoning.

**Strengths:**

* **Simple, drop-in architectural tweak** with clear recipe (head-merge for GLA; dk upscaling for Mamba2), applied before long-context post-training. Results show consistent gains on recall/ICL at the same data budget, e.g., GLA recall average +1.47 points (41.42→45.18) with unchanged parameter count and larger total state (12.48M→18.72M).

**Weaknesses:**

1. **Long-context retrieval is still near-zero at large lengths; claims feel overstated.**
   Despite improvements at 4–32K, performance collapses by 64K (GLA passkey: 0.01; Mamba2 NIAH-Single-2: 0.00). This undermines the central claim of *enhanced long-context recall* “without compromising other capabilities.” The absolute ability at the longest contexts remains effectively absent.

2. **Efficiency claims lack compute evidence.**
   The paper asserts “no or negligible increase” in parameters and *efficiency* via post-training, but provides **no wall-clock, throughput, memory, or latency** measurements. Given Section 3.4 explicitly notes token-mixing cost **scales linearly with state size**, an empirical compute table is needed to support the efficiency narrative.

3. **Novelty claim vs. prior/concurrent state-expansion work is under-substantiated.**
   The paper positions itself as “the first” to expand RNN states *via post-training*, yet concurrently cites MoM/LaCT/low-rank expansion efforts and does not implement **direct baselines** under the same training budget. The empirical comparison is therefore incomplete, and the novelty statement feels strong relative to coverage.

**Questions:**

* **Grammar:** “We **will be released** the model checkpoints…” → “We **will release** the model checkpoints…” (Reproducibility).
* **Duplication:** “we **evaluate also evaluate** with different number of in-context demonstrations” → remove duplicate “evaluate” and fix plural (“numbers”).
* **Equation indexing bug (A.1):** second line should use (Y'^{(l)}), not (Y'^{(l-1)}):
  Current: $(Y^{(l)}= \mathrm{FFN}^{(l)}(Y'^{(l-1)}) + Y'^{(l-1)})$
  Expected: $(Y^{(l)}= \mathrm{FFN}^{(l)}(Y'^{(l)}) + Y'^{(l)})$.
* Common-sense averages slightly **decrease** vs. original models (GLA: 46.74→46.37; Mamba2: 52.93→52.33), so “without compromising” is not universally true.
* No variance/error bars or multi-seed runs; improvements are modest and could be within noise on some tasks.

---

> ### Author Response · Authors · 2025-11-25
> **Response to Reviewer ambX | Part 1 of 2**
>
> Dear Reviewer
>
> We appreciate your time and effort in reviewing our paper. In response to the weaknesses you mentioned, we would like to make the following remarks.
>
> ## Long-context retrieval is still near-zero at large lengths.
>
> We first emphasize that the main performance gains from StateX lies in recall-intensive tasks (Table 2) and in-context learning (Table 3), as presented in the Main Results section. Moreover, **32K context length is very long according to the standards in** **RNN** **literature, since most work focuses on context lengths within 10K**. For instance, GLA and Mamba2 are trained with contexts lengths shorter than 10K.
>
> Improving RNN recall at 64K and beyond is an open problem that is outside the scope of our work, and is orthogonal to our research question, which is focused on improved RNN recall abilities.
>
> ## Comparison with Previous State Expansion Works
>
> We would like to highlight that the previous state expansion works (e.g., MoM [1], LaCT [2] etc.) do not expand the states through post-training. Their models are pre-trained from scratch. Therefore, our work is the first study that expands RNN states via *post-training*.
>
> Nonetheless, we have conducted a comparison with MoM, which is the most relevant model with larger states. To ensure fair comparison, we train an MoM model from scratch using the same data we used for StateX. The training hyperparameters are also identical. Then, we compare StateX with it on common-sense reasoning, recall-intensive tasks, and in-context learning.
>
> ### Results
>
> The results are presented in the following three tables. It is evident that MoM performs much worse than StateX in terms of common-sense reasoning, recall abilities, and in-context learning, when **trained on the same amount of data**. Also, MoM is much slower than StateX (see below for throughput measurements).
>
> **Common-Sense Reasoning**
>
> | **Model**           | **PIQA** | **Hella** | **Wino** | **ARC-e** | **ARC-c** | **SIQA** | **Average** |
> | :------------------ | :------- | :-------- | :------- | :-------- | :-------- | :------- | :---------- |
> | StateX-GLA (ours)   | 69.8     | 37.2      | 54.9     | 53.9      | 22.5      | 40.0     | 46.4        |
> | StateX-Mamba (ours) | 73.7     | 45.1      | 60.0     | 64.0      | 29.6      | 41.6     | 52.3        |
> | MoM                 | 63.3     | 30.4      | 50.8     | 45.2      | 18.8      | 37.4     | 41.0        |
>
> **Recall Abilities**
>
> | **Model**           | **SWDE** | **SQuAD** | **TQA** | **NQ** | **Drop** | **Average** |
> | :------------------ | :------- | :-------- | :------ | :----- | :------- | :---------- |
> | StateX-GLA (ours)   | 50.3     | 59.2      | 55.0    | 21.8   | 39.6     | 45.2        |
> | StateX-Mamba (ours) | 56.2     | 57.9      | 63.7    | 36.4   | 36.4     | 50.1        |
> | MoM                 | 34.4     | 49.6      | 50.1    | 16.0   | 33.9     | 36.8        |
>
> **In-Context Learning**
>
> | **Model**            | **8-shot** | **16-shot** | **24-shot** | **32-shot** | **Average** |
> | :------------------- | :--------- | :---------- | :---------- | :---------- | :---------- |
> | StateX-GLA (ours)    | 48.2       | 52.4        | 52.0        | 50.6        | 50.8        |
> | StateX-Mamba (ours) | 47.7       | 52.3        | 53.0        | 49.8        | 50.7        |
> | MoM                  | 42.6       | 42.2        | 42.9        | 42.3        | 42.5        |
>
> **Takeaways.**
> StateX outperforms MoM on common-sense reasoning, recall-intensive tasks, and in-context learning by a large margin, when trained on the same data.

---

> ### Author Response · Authors · 2025-11-25
> **Response to Reviewer ambX | Part 2 of 2**
>
> ## Efficiency Gains Lack Compute Evidence
>
> We have measured the throughput of the original GLA and Mamba2, their StateX versions, and MoM [1], in training, prefilling (which is also correlated with latency), and decoding.
>
> **Experimental Settings**
>
> The RNN component of each model is implemented with kernels from the widely used flash-linear-attention GitHub repository (4K stars). Inference throughput measurements are performed on one NVIDIA A800-SXM4-80GB GPU, and training throughput is measured on a machine equipped with eight NVIDIA A800-SXM4-80GB GPU. The training framework is implemented with the popular HuggingFace Accelerate framework with data parallelism (which is popular for multi-GPU training).
>
> **Training Efficiency Results**
>
> The following table reports the training throughput (tokens per second) for
>
> | **Model**           | **Vanilla GLA** | **StateX-GLA** | **Vanilla Mamba** | **StateX-Mamba** | **MoM** |
> | :------------------ | :-------------- | :------------- | :---------------- | :--------------- | :------ |
> | Training Throughput | 129.1K          | 122.1K         | 108.5K            | 104.3K           | 55.9K   |
>
> **Inference Efficiency Results**
>
> The following table reports the prefilling throughput (tokens per second) for various context lengths T. For consistency and practicality, we adjust the batch size B such that B*T is constant.
>
> | **Model**               | **1K ctx. len.** | **4K ctx. len.** | **16K ctx. len.** | **64K ctx. len.** | **Average** |
> | :---------------------- | :--------------- | :--------------- | :---------------- | :---------------- | :---------- |
> | *Vanilla GLA*           | *72.5K*          | *72.6K*          | *72.2K*           | *70.4K*           | *71.9K*     |
> | **StateX-GLA (ours)**   | **70.0K**        | **70.1K**        | **69.5K**         | **68.0K**         | **69.4K**   |
> | *Vanilla Mamba*         | *44.0K*          | *44.1K*          | *44.2K*           | *50.7K*           | *45.7K*     |
> | **StateX-Mamba (ours)** | **45.1K**        | **45.3K**        | **42.6K**         | **52.2K**         | **46.3K**   |
> | MoM                     | 20.3K            | 28.1K            | 29.5K             | 31.4K             | 27.3K       |
>
> The following table reports the decoding throughput for various batch sizes. For RNN models, the decoding speed is independent of the context length.
>
> | **Model**               | **BSZ = 64** | BSZ = 128  | **BSZ = 256** | BSZ = 512                | **Average** |
> | :---------------------- | :----------- | ---------- | :------------ | ------------------------ | :---------- |
> | *Vanilla GLA*           | *3548.1*     | 3814.6     | *9594.8*      | 10225.9                  | 6795.9      |
> | **StateX-GLA (ours)**   | **3769.6**   | **6371.1** | **7082.5**    | **7394.9**               | **6154.5**  |
> | *Vanilla Mamba*         | *2275.5*     | 3033.7     | *4754.9*      | 5730.0                   | 3948.5      |
> | **StateX-Mamba (ours)** | **2173.5**   | **3120.6** | **4350.8**    | **4836.8**               | **3620.4**  |
> | MoM                     | 46.3         | 47.5       | 48.2          | ***Out of CUDA Memory*** | 47.3        |
>
> *Note: The average throughput of MoM only accounts for the BSZ of 64, 128, and 256, since MoM is out of CUDA memory (80GB) when BSZ = 512.*
>
> **Takeaway.** StateX versions of GLA and Mamba2 almost as fast as the original models in prefilling and training, and slightly slower in decoding. Compared to MoM, StateX is:
>
> - roughly 1.9x to 2.2x faster in training;
> - roughly 1.7x to 2.5x faster in prefilling;
> - roughly up to 147x faster (using batch size of 256) during decoding, while MoM gets out of CUDA memory in the batch size of 512.
>
> ## Decrease in Common-Sense Results
>
> We emphasize that the baseline to compare with is the LPT model, *not* the original model, because the original model has not undergone long-context training and, thus, has a much shorter context window. **The relative difference between StateX models and LPT models is smaller than 0.5%**, and we think it is reasonable to say that this is a negligible difference.
>
> ## Multi-Seed Runs
>
> While we recognize the importance of multi-seed runs and error bars, we would like to emphasize that, due to resource constraints, **it is uncommon in related literature to report multi-seed runs** **in experiments of this scale****.** For instance, the original papers of GLA and Mamba, as well as the two papers mentioned by the reviewer (MoM and LaCT), do not perform multi-seed runs. The performance gain of StateX is significant by the standards of relevant studies.
>
> ## Various Grammar Mistakes/Typos
>
> Thank you for spotting these mistakes, we will update the equations in A.1, and fix the grammar mistakes mentioned by you.
>
> If you have any concerns left unaddressed, we are eager to engage in further discussions.
>
> Thank you.
>
> **References**
>
> [1] MoM: Linear Sequence Modeling with Mixture-of-Memories.
>
> [2] Test-Time Training Done Right.

---

> ### Author Response · Authors · 2025-11-27
>
> Dear Reviewer ambX,
>
> We genuinely appreciate your time and effort in reviewing our paper.
>
> To address your concerns, we've added a detailed response supported by extensive experiments. We would like to know if you have any other unaddressed concerns/questions and would be glad to have a further discussion. If all your concerns are resolved, we kindly hope it will be reflected with a more favorable rating on our paper.
>
> Thank you,
>
> Authors

---

### Note · Authors · 2025-12-25

**Comment:**

After careful consideration, all authors have collectively agreed to withdraw this paper (StateX: Enhancing RNN Recall via Post-training State Expansion, Submission Number: 20019).

**Withdrawal Confirmation:**

I have read and agree with the venue's withdrawal policy on behalf of myself and my co-authors.